# Decapod-inspired pigment modulation for active building facades

Raphael Kay [1,2,3] ✉, Charlie Katrycz[1], Kevin Nitièma[3], J. Alstan Jakubiec[3,4] & Benjamin D. Hatton [1] ✉

Typical buildings are static structures, unable to adjust to dynamic temperature and daylight fluctuations. Adaptive facades that are responsive to these unsteady solar conditions can substantially reduce operational energy inefficiencies, indoor heating, cooling, and lighting costs, as well as greenhouse-gas emissions. Inspired by marine organisms that disperse pigments within their skin, we propose an adaptive building interface that uses reversible fluid injections to tune optical transmission. Pigmented fluids with tunable morphologies are reversibly injected and withdrawn from confined layers, achieving locally-adjustable shading and interior solar exposure. Multicell arrays tiled across large areas enable differential and dynamic building responses, demonstrated using both experimental and simulated approaches. Fluidic reconfigurations can find optimal states over time to reduce heating, cooling, and lighting energy in our models by over 30% compared to current available electrochromic technologies.

[1] Department of Materials Science and Engineering, University of Toronto, Toronto, ON M5S 3E4, Canada. [2] Department of Mechanical and Industrial Engineering, University of Toronto, Toronto, ON M5S 3G8, Canada. [3] John H. Daniels Faculty of Architecture, Landscape and Design, University of Toronto, Toronto, ON M5S 2J5, Canada. [4] School of the Environment, University of Toronto, 149 College Street, Toronto, ON M5T 1P5, Canada. ✉email: raphael.kay@mail.utoronto.ca; benjamin.hatton@utoronto.ca

Biological organisms have evolved a vast collection of dynamic regulatory controls to maintain an equilibrium with their environment. As a remarkable example, Antarctic krill, *E. superba*, can actively change color within minutes depending on sunlight intensity for ultraviolet protection[1]. Like several decapods[2–7], krill store and disperse pigment throughout the cells within their skin, evolving a rapid and reversible response mechanism for solar shading (Fig. 1a–d)[1,8,9].

Buildings, in contrast, are generally unequipped to achieve adaptive solar shading responses, built with static outer facades, despite operating within highly variable temperature and light regimes[10–12]. A skyscraper in a typical seasonal climate, for instance, might experience fluctuations in solar radiation from almost 0 to an astounding 800 W/m² within a day. Static glazing materials cannot regulate optical transmission in response to these fluctuating solar loads[13]. Unshaded windows, for instance, allow excessive solar heating in the summer, contributing to high seasonal cooling energy costs[10–12]. Windows with near-infrared-reflective and low-emissivity coatings, on the other hand, limit crucial solar ingress in the winter, and incur equally-consequential seasonal heating energy costs[11]. Beyond heating effects, windows also must provide sufficient total interior illumination while limiting excessive localized glare[14]. Today, in large part because outer glazing materials cannot adaptively or locally shade against solar loads, buildings consume almost 75% of the U.S. national electricity supply[15] and approximately one third of the global energy supply[16]. Adaptive glazing materials, capable of both dynamic and localized solar shading, have the potential to significantly improve energy efficiency for a recognizable impact on climate change[14,17], and could additionally improve indoor human comfort[18,19].

Despite this potential impact[17,20], active shading in buildings has been difficult to achieve. An ideal optically-active building facade should be locally-responsive (maximize light transmission, but limit glare), digitally-controllable (optimize material properties and building configuration), low-cost and scalable across large areas, while also energy-efficient to operate. Currently, many buildings only achieve shading through manual, large-scale, mechanical blinds[21,22]. Rotating shading frits[23] and other automated mechanical structures[24,25] have also been tested. However, these macroscale mechanical approaches often rely on fixed rigid movements and have low spatial resolution[10,26] for localized shading responses[21,22]. Certain smart materials have also been developed for active shading, but have practical limitations. Electrochromic systems, for instance, which use chemical redox reactions to control optical transmission, are expensive (with the latest prices reported around $100–500 USD/m²)[27–31], complex to manufacture[28,29,32], and typically reliant on energy-intensive sputter-deposition processes[33], restricting market viability[12,27–29,34]. More experimental chromogenic systems that use electrically-reorientable liquid-crystals[35–37] and suspended-particles[38,39], as well as active polymers that leverage dielectric elastomer actuations[40–42], require a continuous energy supply to maintain a bleached state. Finally, stimulus-responsive materials and actuators (e.g., photochromics[12,28,35,43–45], thermochromics[12,28,43–47], and hygroscopics[48,49]) suffer functional restrictions of their own[28,41], and cannot be digitally controlled or decoupled from their unique environmental triggers, limiting both the capacity for digital information processing and tunable user control.

In contrast, biological organisms often leverage tissue-scale fluid and soft material mechanisms to regulate interfacial properties within evolving environments. Mammals dilate blood vessels near their skin to control rates of convective heat loss[50–53]; cephalopods stretch pigment-containing sacs to generate colorful displays for adaptive camouflage and visual communication[54–59]; brittle stars transport fluidic cells between sub-surface regions to

regulate photoreception[60]; and decapods (e.g., krill, crab) move pigments within their skin[2–7] to thermoregulate and dynamically shade against the sun[1,61]. In low-light conditions, krill store pigments in a central reservoir within sub-surface chromatophore cells. Under intense light exposure, they then quickly (<20 min) spread pigment through the radially-branching microtubules of the chromatophore, expanding the diameter of pigment coverage from <100 μm within the reservoir to >500 μm when expanded across the cell (Fig. 1a, b). In aggregate, this extended pigment coverage significantly changes the optical appearance of the skin (Fig. 1a, b)[1]. Crucially, only a small volume of pigment is required to actively and efficiently shade a large surface region, expanding from a point (reservoir) to an area.

We hypothesize that this intracellular actuation of confined pigment, scaled up as a material layer within a building facade, can replicate the dynamic optical response of biological tissue (Fig. 1g–j). Here, we combine principles of microfluidics, self-organization, and digital actuation to conceptualize a large area building interface that can differentially sense and react to local optical conditions. We show that pigment-containing fluids, confined within layered devices, can be injected and withdrawn (Fig. 2d) to control color and shading, interior light intensity, and temperature. The self-organizing morphology of these injected fluids is controlled through the non-equilibrium dynamics of branching instabilities. This mechanism allows us to demonstrate that building facades with adaptive and reversible fluidic shading can achieve significant improvements to energy efficiency.

## Results

**Characteristic branching morphologies of injected fluids.** Marine organisms use radially-branching vascular networks to disperse pigment within their skin. Branched area coverage increases the effective scale of pigment dispersion, where a small volume of pigment fluid can expand across a much larger surface area in a branched morphology compared to a uniform disk. Here, we generate radially-branching pigment fluid morphologies within layered devices for tunable optical transmission. We demonstrate reversible pigment coverage, from a point reservoir to a large area, analogous to pigment dispersal for optical control in the krill chromatophore cell (Fig. 1f, g).

We control branched morphologies through the viscous fingering (VF) effect (Fig. 2a–e, Supplementary Movie 1). VF is a well-known mechanism for branched pattern formation, where interfacial instabilities grow as a less viscous fluid is forced under pressure into a more viscous fluid, while confined between two closely spaced plates[62]. This patterning has been widely demonstrated and characterized using the quasi-2D Hele-Shaw (H-S) cell[63–68], where fluid parameters and cell geometry can be controlled to tune the morphology and planar area fraction of the invading fluid.

Within H-S cells, injection flow rate affects pressure at the interface between the two fluids, and can be controlled experimentally. At sufficient flow rates, this interface can expand in a budding and branching pattern, as the guest fluid bifurcates to form fingers within the host fluid. The curvature of this interface, if unstable, is locally amplified. For vertical H-S cells, with density-matched fluids (no buoyant forces, see[62]), the unstable tip-splitting growth of fingers occurs when an amplification factor of a specific finger width, $a_\lambda > 0$, for

$$a_\lambda = 3V\Delta n - \sigma\left(\frac{\pi b}{\lambda}\right)^2 \tag{1}$$

Where $\Delta n = n_h - n_g$, $n_h$ is the host fluid viscosity, $n_g$ is the invading guest fluid viscosity, $V$ is the interfacial velocity, $b$ is the gap height between plates, $\sigma$ is the interfacial surface tension, and $\lambda$ is the finger width, or wavelength, of the instability[62]. Unstable

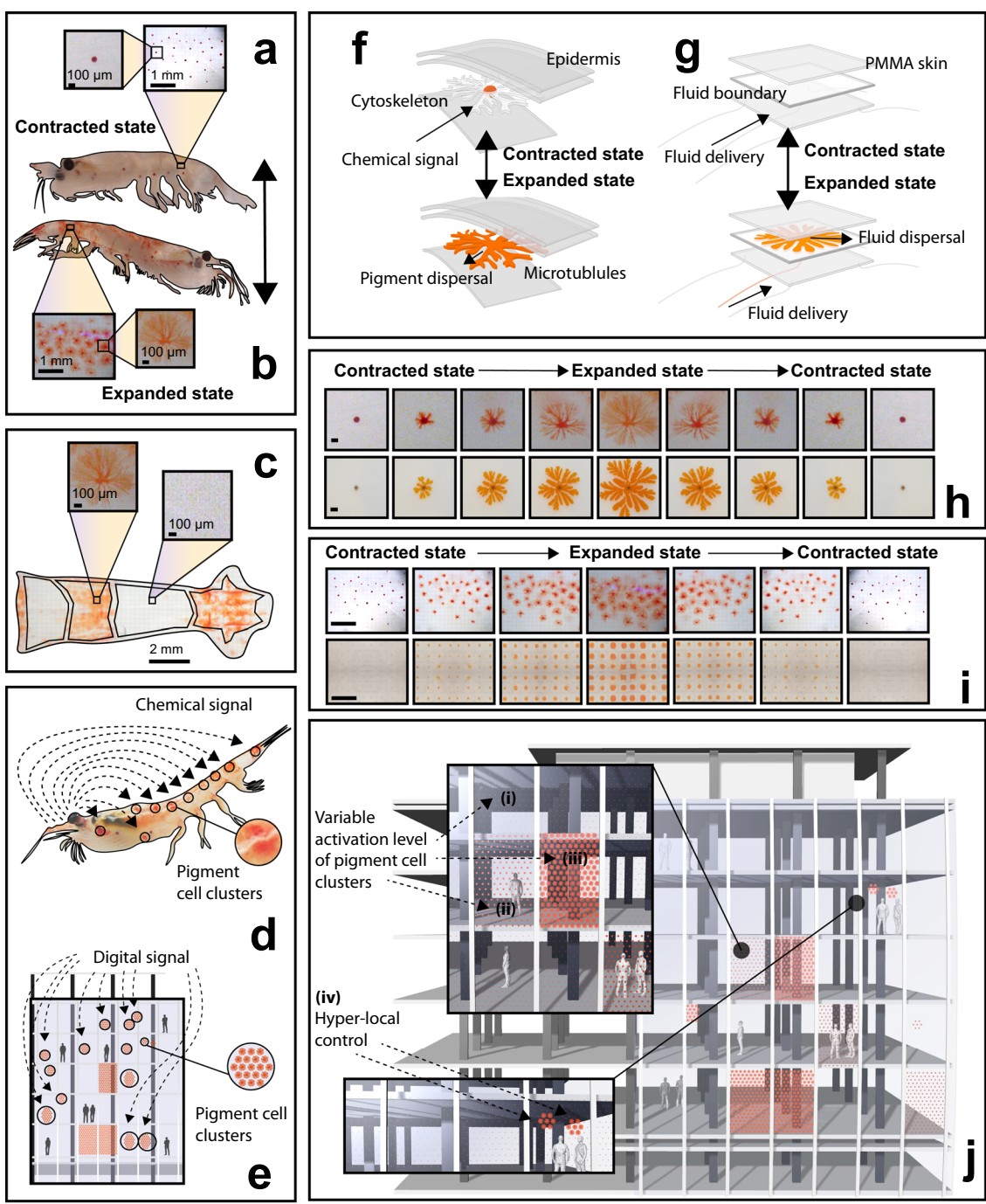

**Fig. 1 Biological inspiration for active pigment dispersal in buildings. a**, **b** Reversible chromatophore activation in male Antarctic Krill when unexposed (**a**) and exposed to light (**b**) as a mechanism for dynamic solar shading. **c** Localized chromatophore coverage of the abdominal segments in Krill. **d**, **e** Schematic comparing the activation pathways for both biological chromatophore clusters in Krill (**d**) and synthetic chromatophore clusters in buildings (**e**), to control the ingress of solar radiation through the skin of Krill (**d**) and facade of buildings (**e**). **f**, **g** Exploded perspectival cross-section showing both the contracted (top) and expanded (bottom) state of a single chromatophore in both Krill (**f**) and synthetic device (**g**). **h** Images comparing complete chromatophore expansion and contraction sequence in both Krill (top) and synthetic device (bottom). Top scale bar is 100 μm. Bottom scale bar is 2 cm. (**i**) Images comparing complete expansion and contraction sequence for a cluster of chromatophores in both Krill (top) and synthetic device (bottom). Top scale bar is 1 mm. Bottom scale bar is 5 cm. Bottom image is stitched from four images of a 4 × 3 pixel array. **j** Render showing dynamic and localized synthetic chromatophore activation within a building facade, where multiple activation states (i, ii, iii) and hyper-local control (iv) can be achieved. All images of Antarctic Krill in a-d, h-i were provided by Lutz Auerswald.

branching tips will grow if the growth rate (left hand term) is large enough to overcome the smoothing-effect of surface tension on the decay rate (right hand term). However, if the direction of flow is reversed, we can expect the reversal of stability. The sign of $a_\lambda$ changes, due to the change in flow direction (swapping the host and guest fluid viscosities), causing a net decay of finger amplitudes. This mechanism of stability reversal allows our design to be flow reversible: when driven in the forward direction (pigment injection), instabilities cause branching morphologies. When driven backwards (pigment withdrawal), the curvature-

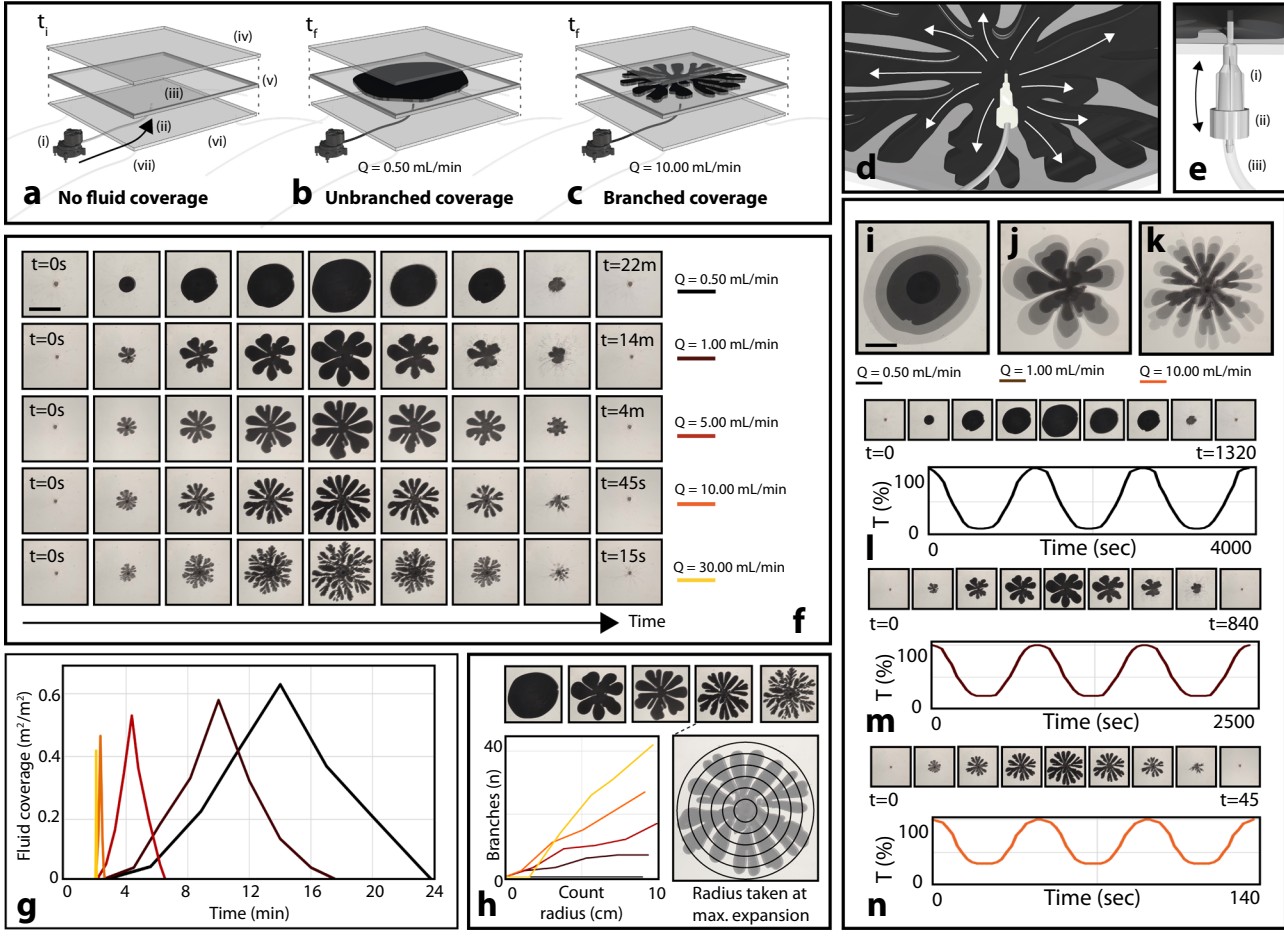

**Fig. 2 Tuning shading coverage by actively controlling pigment morphology. a–c** Schematic of a single fluidic cell, with no pigment fluid coverage (**a**), unbranched pigment fluid coverage (**b**), and branched pigment fluid coverage (**c**). System components: (i) digitally controlled peristaltic pump; (ii) inward fluidic pigment flow with pressure; (iii) active fluidic pigment layer, 1 mm thick; (iv) first rigid plate; (v) outer gasket; (vi) second rigid plate; (vii) drain tubing for temporary fluid displacement. **d** Demonstrating the branching of pigment fluid when introduced at higher speed. **e** Inlet design: (i) needle; (ii) luer connector; (iii) hose connector. **f** Reversible pigment injection/withdrawal, where degree of branching is determined by injection flow rate (Q) from 0.5–30 mL/min. Scale bar is 15 cm. **g** Fluid coverage as a function of time for experiments pictured in (**f**). **h** The number of fluidic branches increases with flow rate. All measurements taken once pigment fluid fully dispersed. **i–k** Overlayed images of pigment fluid dispersal over time for three different flow rates. Scale bar is 5 cm. **l–n** Cyclical light transmission measurements across three pigment fluid dispersal and contraction cycles. Images represent first cycle. Light intensity varies for each system at full actuation. Because maximum light intensity was set to 100 lux, plots in (**l–n**) additionally display relative measured interior light intensity behind each cell in lux.

dampening effect allows for the coordinated collapse of the branched fluid network back to its source.

The width of the branches in the pattern is a critical design parameter that helps to control pigment fluid area fraction and, subsequently, light transmission. The critical finger width $\lambda_c$ divides stable and unstable growth,

$$\lambda_c = \pi b \sqrt{\frac{\sigma}{3V\Delta n}} \quad (2)$$

Wavelengths less than $\lambda_c$ are stabilized due to the increase in the decay term in Eq. 1. All wavelengths longer than $\lambda_c$ are unstable, with a practical upper limit at the longest wavelength that can fit along the interface, determined by the radius of the fluid injection (Supplementary Fig. 8). The most accelerated instability that is characteristic of the branched pattern, is characteristic wavelength,

$$\lambda^* = \sqrt{3}\lambda_c = \pi b \sqrt{\frac{\sigma}{V\Delta n}} \quad (3)$$

We developed H-S cells ($30 \times 30 \times 0.1$ cm$^3$) using PMMA sheets (fabrication details in SI), and controlled reversible fluid injections through a central port and digital syringe pump (design partially adapted from REF)[69]. If we assume a stably expanding circular fluid disk between consistently-spaced rigid plates, the velocity of the fluid interface is linearly proportional to the injection flow rate, $V = \frac{Q}{2\pi r b}$ (derivation in SI). Therefore, injection flow rate (Q) can be used to modulate V, and the onset of branching instability ($\lambda^*$). Experimentally, Q is highly practical for establishing control over pigment morphology due to its digital tunability.

Based on Eq. 1, we chose two immiscible fluids – a transparent mineral oil (288 cP, 20 °C) and aqueous carbon suspension (0.89 cP, 20 °C) – as the host and guest phase, respectively. Low injection rates (e.g., 0.5 mL/min) of the aqueous pigment phase corresponded to patterns of decreased branching during dispersal (Fig. 2f, top row), while increasing flow rate increased branching (Fig. 2f, moving downwards, Fig. 2h). For a consistent dispersal radius within a H-S cell, differences in branching resulting from injection flow rate corresponded to disparities of up to 21% in pigment fluid area coverage, from 42% coverage for a highly-branched state (Fig. 2f, top row), up to a maximum of 63% coverage for a non-branching state (Fig. 2f, bottom row, Fig. 2h). The well-established linear relationship between characteristic wavelength and interfacial velocity[62] is confirmed using Eq. 3 (Supplementary Fig. 5c). Systematic control of $\Delta n$ (and $n_{guest}/n_{host}$)

was also demonstrated to tune area fraction by 25% (for the same relative pigment fluid radius), from 45% to 70% coverage (Supplementary Fig. 3b). Other results demonstrating geometric control over viscous fingering to tune optical transmission are available in Supplementary Fig. 1–7.

**Reversibility of injections and switchable injection stability**. To avoid buoyant forces in vertically-oriented cells, the aqueous pigment phase was adjusted (by adding ethanol) to an equivalent density such that $\rho_a - \rho_o = 0$, where $\rho_a, \rho_o$ are the densities of the aqueous and oil phases, respectively. To control conditions for branching, we assume that the largest wavelength that can be supported on a circle is $\lambda = \frac{c}{2} = \pi r$, where $c$ is the circumference and $r$ its radius. This minimal instability simply transforms a circle into an ellipse, with two crests (fingers) oriented on opposite poles of the major axis. Unstable branching growth will therefore occur for $\lambda_c < \lambda < \pi r$. Stable, non-branching fluid injections ($a_\lambda < 0$) occur when $\lambda_c = \pi r$, and can be accomplished within a vertical cell (Supplementary Movie 4) by ensuring $3V\Delta n < \sigma\left(\frac{b}{r}\right)^2$, where $r$ is the effective radius of the pigment injection within a H-S cell. Plotting $\sigma, \triangle n$ as in Supplementary Fig. 9, a binary phase space is shown, where $a_\lambda = 0$ defines a line through the origin, of slope $m = 3V\left(\frac{r}{b}\right)^2$, separating stable and unstable regions (Supplementary Fig. 9). For a H-S cell aspect ratio $\frac{b}{r}$, one can then control the degree of instability using any of $\sigma, \triangle n, V$ (Supplementary Figs. 8–9). We found that less branched morphologies were often better able to retract due to pinching effects in narrower fingers. Non-branched morphologies were found to be repeatable (injection/withdrawal) after 100 cycles.

**Reversible injection tuning for dynamic shading**. With well-defined control over branching stability, pigment fluid morphology, and pigment fluid reversibility, we demonstrated reversible, programmable pigment fluid injection to tune optical transmission and shading in H-S cells. In their transmissive (clear) state, cells contained a transparent fluid (mineral oil, 288 cP), enabling full transmission of visible light (Fig. 3b). To shade, we injected a less viscous pigment fluid (carbon black suspension in water-glycerol solution, 0.89–288 cP) into the mineral oil layer (Figs. 1h–i, 2a, d–e). We measured optical transmission for these aqueous carbon suspensions and found a minimum concentration (0.02 g C/mL H$_2$O) for zero light transmission (300–3400 nm) for a cell thickness of 4 mm (Supplementary Fig. 10, Fig. 3b). We measured light transmission behind a cell across a complete dispersal and retraction sequence. As expected, transmitted light decreased as a function of pigment fluid area (Supplementary Fig. 7), to decrease interior visible light intensity by 91%, 80% and 67% for maximum injections with flow rates of 0.5, 1.0, and 10.0 mL/min, respectively (Fig. 2l–n, respectively). We can therefore use flow rate to control branching and relative area coverage, modulating light transmission through the cell by 24% for differentially-branched patterns of the same maximum radius (Supplementary Movie 2, Supplementary Figs. 3a, 4a). Additionally, by controlling the branching effects of the pattern with viscosity differences (ratios), we modulated light transmission by 12% for patterns of the same radius (Supplementary Movie 3, Supplementary Figs. 1a–b, 2a–b, 3b, 4b).

**Proportional fluidic optical and thermal responsiveness**. Radiative heat transfer through a building facade is a major contributor to operational cooling and heating costs[14,70]. Transmitted light through a facade, and the energy that is absorbed and reemitted as heat into or out of a building, must be appropriately regulated. In buildings, the fraction of solar radiation that is transmitted into a building is captured by a solar heat gain coefficient, $SHGC = T_{sol} + A_{sol} \cdot N$, where $T_{sol} = \left\{\sum_{\lambda=100nm}^{1mm} T_{\%}\lambda\right\}$ is the transmission fraction across the solar radiation spectrum on Earth, $A_{sol}$ is the solar absorptance fraction, and $N$ is the inward reemission fraction[71]. Decapods, like the sand fiddler crab, control radiative solar heat gain to thermoregulate by managing the volume of pigment dispersed within their chromatophores[61]. Analogously, our fluidic building layers can achieve a variable response to incident light by managing the amount of pigment fluid distributed over their cell areas.

To demonstrate this adaptive and proportional response, we fabricated a multicell facade with 16 independent injection sites, each with a local photosensor and thermocouple (Fig. 3a, c). Using the photosensor input behind each cell, a digital negative feedback system was developed for the pigment fluid to maintain a light transmission setpoint, given variable incident light intensity (Fig. 3g–j). An optical stimulus of 100 lux was directed at each cell (Fig. 3g), triggering a temporary and proportionate response (Fig. 3h). 20 mL of pigment was injected (10 mL/min) across 115 s to shade the sensor and restore optical transmission to a set value of 100 lux (Fig. 3i).

For analogous temperature-driven control, we placed a thermocouple on a PMMA sheet 3 cm behind each cell to control a pigment injection response to temperature (Fig. 3k–n). Visible and infrared (IR) light transmission through the cell from an applied heat source elevated the sensor temperature from 22 °C to 38 °C, and triggered a 20 mL pigment injection (10 mL/min) within 115 s to shade the sensor. The measured temperature of the uninsulated acrylic sheet returned to 22 °C after 16 min (Fig. 3m), demonstrating a thermoregulatory effect governed by optical properties, and generally independent of the thermal conductivity of a building facade.

**Spatially-differential optical responsiveness**. Crucial in the camouflaging, shading, and thermoregulatory efforts of several marine organisms is the coordinated differential response of independent fluidic cells across the skin. In buildings, this localized shading control might be similarly beneficial, where spatially-differentiated shading responses could provide glare control without sacrificing diffuse light transmission, and provide desirable differences in daylight penetration across a large space[18,19]. We mimicked this local actuation capacity in biology to regulate spatially-varied light transmission in building facades. Regions across a multicell facade were individually illuminated (+100 lux), in a sequential manner, and each responded within 15 s (Fig. 4a–b, Supplementary Movie 5). With reversibility and pattern stability in mind, we injected pigment slow enough to avoid branching instability. The resulting stable circular patterns also enabled maximum control over area fraction (0% up to a theoretical maximum of π/4, or 79%). A similar response was demonstrated post-illumination, as pigment cells contracted to return light transmission to a preset threshold (100 lux). Regions were also differentially illuminated, i.e., as a light intensity gradient, and each of the 16 independent cells responded proportionally in under 100 s (Fig. 4e, Supplementary Movie 7, Supplementary Fig. 11, experimental setup demonstrated in Fig. 4g), varying a fluidic response between 0–20 mL. The capacity for differential pigment injections across multiple cells over multiple cycles was also demonstrated in Fig. 4d, and in Supplementary Movie 6.

We additionally highlight the possibility for large-area pattern control, generating differential pigmentary responses (Fig. 5b, c) through spatial or injection volume modulation, to match the pixels of large digitized, optofluidic displays (simulated in Fig. 5d). This halftone effect (analogous to screen printing), where a pixel

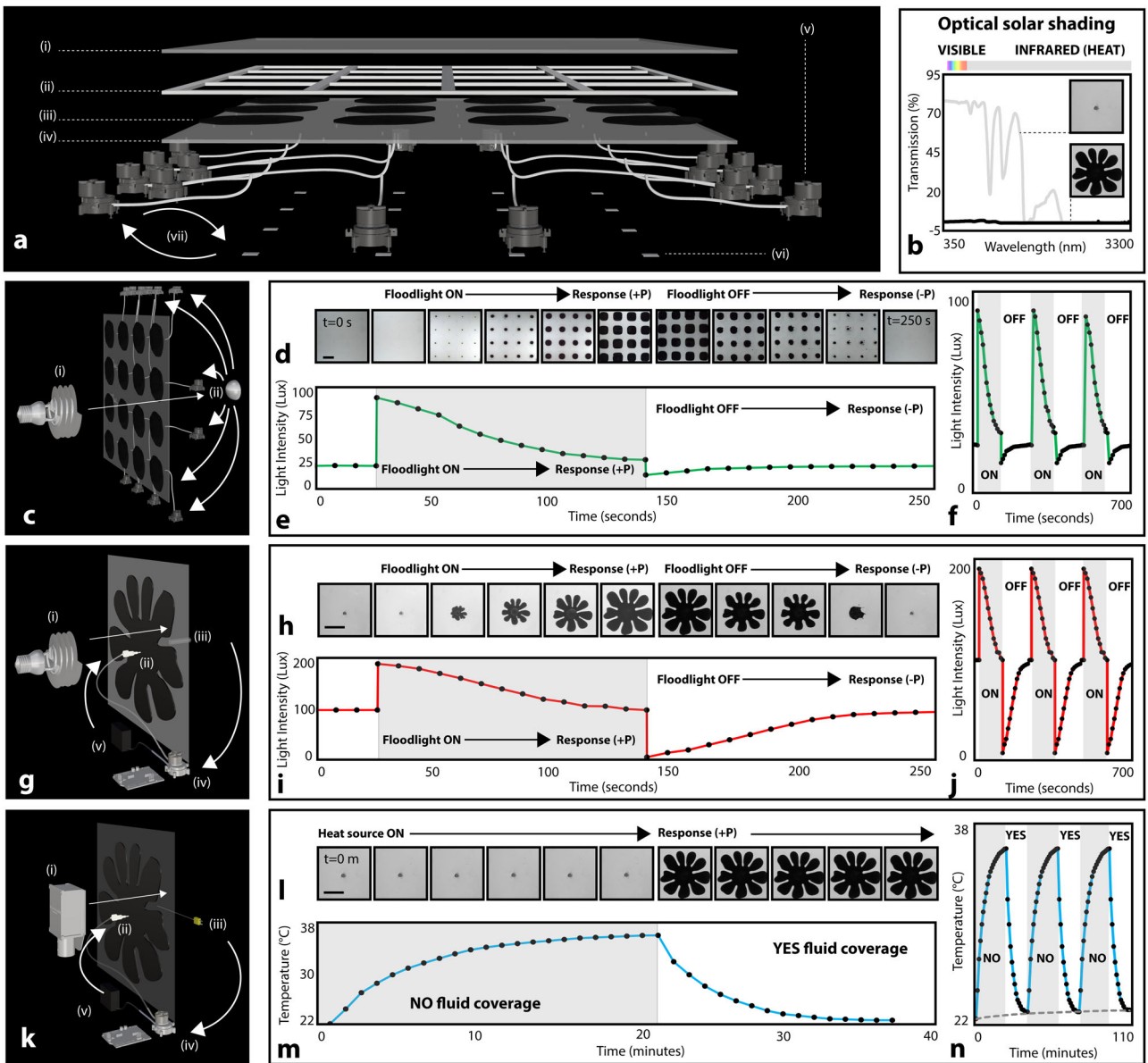

**Fig. 3 Optical and thermal characterization of responsive pigment dispersal for single and multicell system. a** Multicell device: (i) outer expandable layer; (ii) adhesive and gasket; (iii) fluidic pigment layer; (iv) inner rigid plate; (v) digitally-driven peristaltic pump; (vi) light intensity sensor; (vii) feedback loop between sensor and pump. **b** Optical spectrum for pigment fluid (aqueous carbon black, black line) and castor oil (grey line). **c** Experimental setup for data in (**d–f**): (i) light source; (ii) single light sensor. **d** Images showing dispersal and contraction sequence for multicell devices as a response to light. Scale bar is 10 cm. **e** Light intensity as a function of time for single sequence in (**d**). **f** Three sequences of (**e**) to demonstrate consistency. **g** Experimental setup for data in (**h–j**): (i) light source; (ii) dispersed pigment fluid layer; (iii) light sensor; (iv) signal to digital peristaltic pump; (v) control over pigment fluid dispersal. **h** Images showing dispersal and contraction sequence for single-cell device as a response to light. Scale bar is 15 cm. **i** Light intensity as a function of time for single sequence in (**h**). **j** Three sequences of (**i**) to demonstrate consistency. **k** Experimental setup for data in (**l–n**): (i) heat source; (ii) dispersed pigment fluid layer; (iii) thermocouple measuring interior plate; (iv) signal to digital peristaltic pump; (v) control over pigment fluid dispersal. **l** Images showing dispersal and contraction sequence for single-cell facade as a response to temperature. Scale bar is 15 cm. **m** Temperature as a function of time for single sequence in (**l**). **n** Three sequences of (**m**) to demonstrate consistency. Grey line represents control curve, where pigment fluid is dispersed and maintained statically across all three cycles.

array with a varied radius or morphology can create the appearance of a gradient, is digitally achieved with high accuracy for a resolution of $40 \times 40$ cells (approximately $12 \times 12 \, m^2$ for 30 cm devices). Increased spatial resolution can be achieved by varying not only the injection pigment radius, but also the branched morphology. Control of pigment branching allows spatially-programmable variation in area coverage for finer spatial resolutions than a series of circular half-tone pixels can provide.

**Simulated building performance.** Digitally-controlled, dynamic fluid interfaces enable a continuous search for optimal facade configurations and building operational energy efficiency. To assess the performance impact of our fluidic device, we leveraged a well-established building energy modelling tool (EnergyPlus) to estimate the annual energy required to heat, cool, and light a conventional commercial space located in Toronto, Canada (a CAD model is shown in Fig. 6a and more details are described in[72]). Within the EnergyPlus model, heating and cooling loads

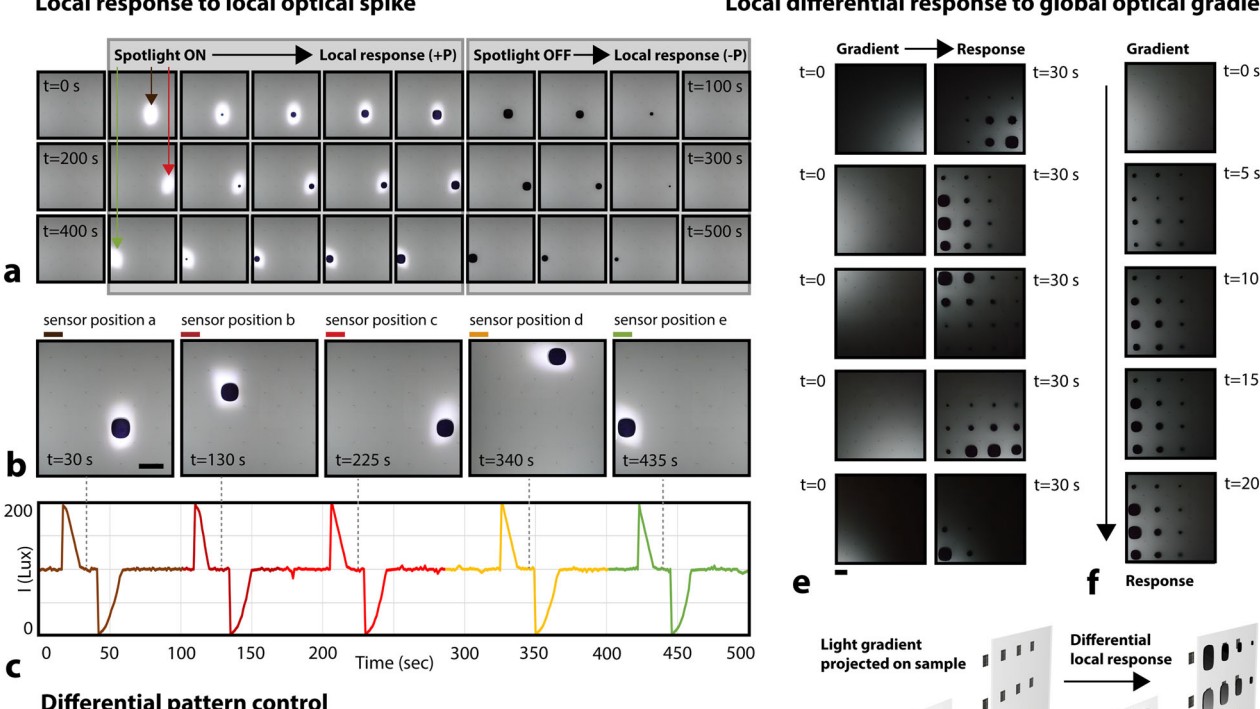

**Fig. 4 Differential pigment responsiveness. a** Independent sequential pigment fluid dispersal/retraction cycles as a response to measured local light intensity behind each cell. Measured value of light intensity drives negative feedback response for each digitally-driven pump. **b** Image captures from Supplementary Movie 5, showing localized responses in five independent positions across the facade over time. Scale bars are 10 cm. **c** Light intensity as a function of time for five different sensors. **d** Image captures from Supplementary Movie 6 to demonstrate differential pattern control over time. Scale bar is 10 cm. **e** Differential pigment fluid response across all sixteen cells for five independent sequences. Scale bar is 7 cm. **f** Second sequence from (**e**). **g** Schematic of experiment in (**e**, **f**).

were calculated using the conduction finite difference solution algorithm[73,74], based on typical climate data[75] on solar flux, sun position, cloud cover, outdoor temperature, and on input data accounting for the conductivity, emissivity, heat capacity, reflectivity, and absorptivity of all opaque and translucent building materials. We compared annual energy usage for conditioning this modelled space when clad along its south facade with (i) a static double-glazed window (control), (ii) our fluidic facade, (iii) an electrochromic (EC) window, (iv) and a dynamic roller shade (RS) interior to the control window.

We leveraged a standard control algorithm that we previously developed[76] to simulate the operation of each dynamic system (ii–iv). The algorithm functions as a naïve energy minimizer, restricted to consistently maintain an illuminance of 300 lux (minimum sufficient daylighting) across half of the floor area, and limit an illuminance of 3000 lux (overlighting correlated with an increased likelihood of visual glare) to 10% of the floor area. Our fluid window was modelled to switch between eight possible states, derived from the effective solar transmission spectra for our layers with a pigment fluid injection fraction between 0–70% (at 10% steps), within the area fraction limit we observed experimentally (Fig. 6e, green). The EC window was modelled to switch between four optical states (transmission spectra correspond to those of a real market product, and are shown in Fig. 6b, black). The RS was modelled to switch between two optical states (up and down), where the up state corresponded to the transmission spectra of the double-pane control window by

itself, and the down state corresponded to the transmission spectra of the roller shade and control window (Fig. 6b, red). To accurately account for building integration, the fluidic and EC systems were modelled on the exterior, while the RS was modelled on the interior, of the standard control window. This double-glazed control window had a conductive heat transfer coefficient (U-value) of 1.81 W/m²K, a visible transmittance of 0.81, and a SHGC of 0.71. Additional zone details, temperature setpoints, and material properties for our simulation are described in Methods.

In simulation, we found that the dynamic operation of our fluid cell reduced total annual operational energy usage within our commercial space (the sum of heating, cooling, and electric lighting energy usage) by 30.1%, 22.9%, and 20.4% compared to EC, RS, and static double-glazing systems, respectively (Fig. 6j). Because tunable pigment injections allow for significant variability in optical transmission (Fig. 6e), our fluidic device can balance heating, cooling, and lighting loads much more efficiently than any of the other dynamic or static systems. In the winter, for instance, the fluid system can transmit significantly more solar energy than an EC window, leading to large comparative heating energy savings (Fig. 6g). In the summer, on the other hand, our fluid system can block more solar energy, while still achieving minimum daylighting, than a static window or RS system, leading to cooling and electric lighting energy savings (Fig. 6h–i). While the static window performance is included as a baseline control, it is not subject to the same spatial daylighting constraints that each of the dynamic systems must meet. In energetic terms, it is

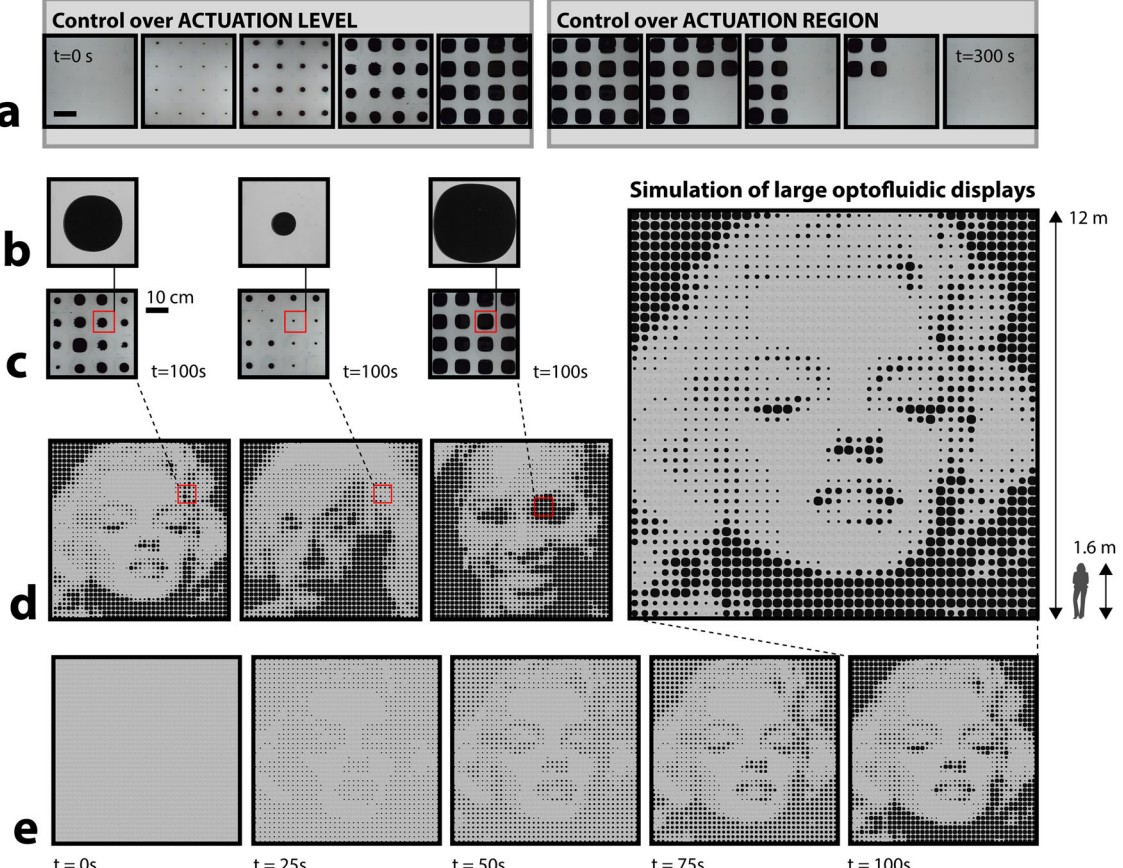

**Fig. 5 Envisioned large-area pigment configurations to display halftone imagery. a** Demonstrated control over pigment fluid actuation level and region. Scale bar is 10 cm. **b-d** Pattern control at three scales over time. **b-c** are physically demonstrated; d is simulated. **e** Pattern definition clarity improves over time, as pigment fluid is dispersed differentially. Original images are from[85-88]. Images of Marylin Monroe and Serena Williams were both licensed as an Image for Artistic Reference from Alamy.

therefore more informative to compare performance strictly between dynamic systems.

**Technoeconomic feasibility**. Our responsive fluidic device can achieve large-area spatiotemporal optical control with low material cost, low embodied energy, simple material processing, and low operational power supply. To assess and quantify the technoeconomic feasibility of our system, we estimated the comparative (i) annual operational energy cost and (ii) material, processing, and fabrication costs of both a state-of-the-art EC window and our active fluidic alternative.

To compare the energy costs for operation, we first estimated the energy required to optimally operate our fluidic facade in simulation across the year. We calculated the total operational energy as the energy for achieving a complete fluid injection/dispersal sequence within a ~0.25 m² panel using our peristaltic pump (5 W · 12 s), multiplied by the number of 0.25 m² panels required to glaze the 9.5 m² south-facing surface (38), multiplied by the number of injection/retraction events that occur annually in simulation (884), or about 1.4 kWh (0.15 kWh/m²). This energy cost is less than 0.25% of the total heating, cooling, and lighting energy savings by our fluid layer over an EC window (578.8 kWh) and is only 30% of a typical operational energy cost estimate of an EC window (0.5 kWh/m²)[17,77]. Furthermore, considering a temporal shading rate of 15.6% (i.e., 1364 h from simulation / 8760 h), we estimate that other chromogenic systems which require a constant low power supply (~5 W/m²) to shade (e.g., liquid-crystal devices[35-37] and suspended-particle

devices[38,39]) might use around 6.82 kWh/m², or about 45 times the operational energy cost of our system.

When considering material, processing, and fabrication energy costs, our trilayer fluid system is relatively inexpensive, consisting of PMMA sheets, common fluids (castor oil, water, ethanol, carbon black), and basic fluid control (mini-peristaltic pump and controller). We suggest a total cost of $40 USD/m² (~$80 USD/m² with a windowpane). EC systems are fabricated with multilayer sputter deposition, and, as redox cells, consist of transparent conductors, electrolytes, and counter electrodes[28]. Costs of EC devices have decreased substantially, but the most recent reported costs from 2010 are still generally between $100–500 USD/m²[2,27–31], and ECs are susceptible to UV degradation. The production and material footprint of EC glazing is also high, where device manufacturing has energy costs 25% greater than those for a double-glazed, argon-filled unit[17].

Using the cost of electricity in Toronto, Canada ($0.893 USD/kWh, 2021), and our system's modelled annual energy performance (343 kWh/year saved compared to a static window, accounting for operational costs), we estimate that the cost of implementing our fluid layer along the south facade of our reference model ($40 USD/m² or $380 USD across 9.5 m²) would pay itself back within 15 months (payback period of 1.24 years).

**Discussion**

Inspired by the dynamic pigment shading response in marine decapods, we demonstrate a low-cost, large-area shading mechanism, leveraging the reversible redistribution of pigment

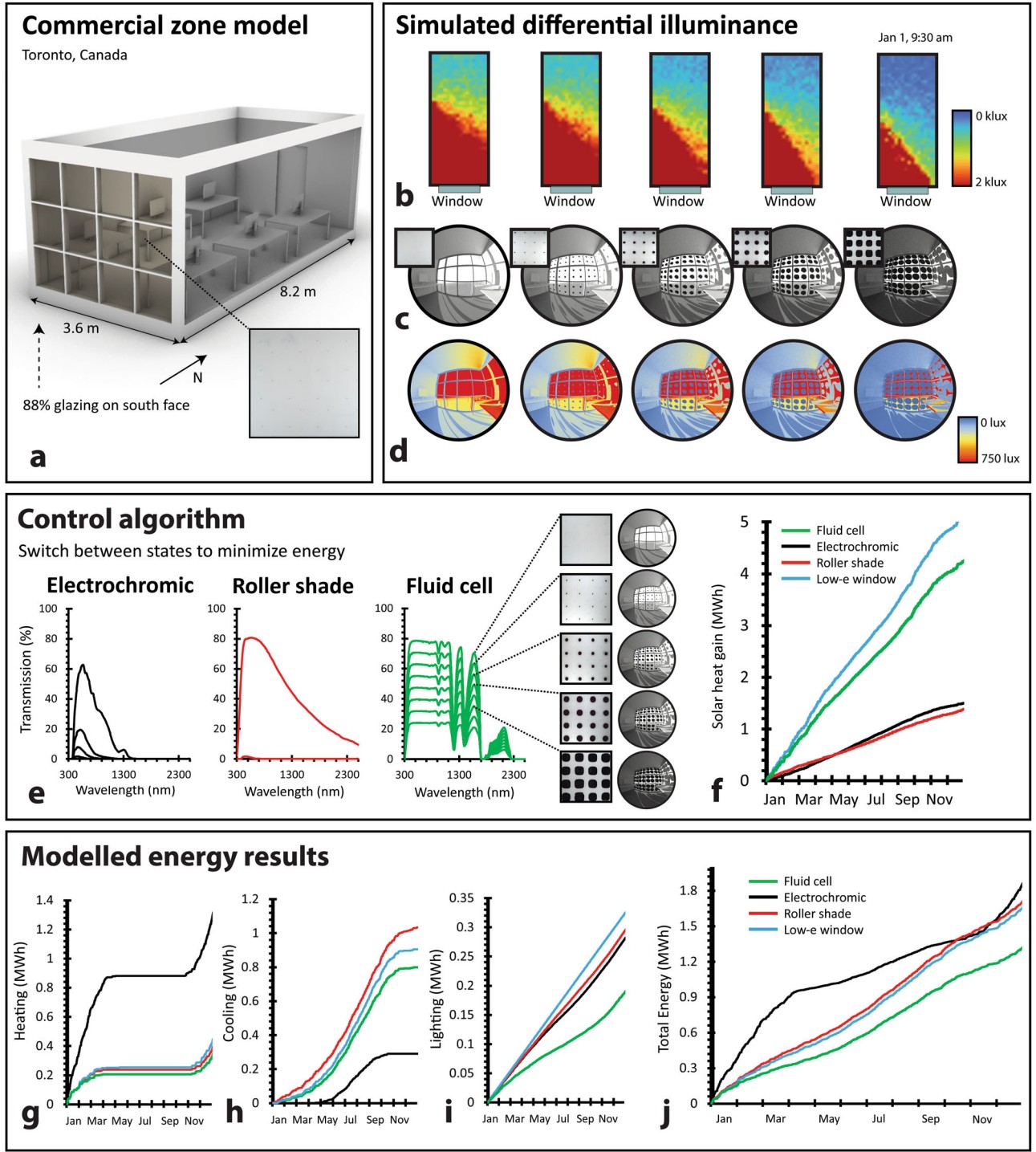

**Fig. 6 Simulated system performance and comparative annual energy savings. a** Commercial zone model for energy simulations, with active south-facing facade. Roof and wall elements were modelled as opaque within the simulation, and appear translucent only for illustration purposes. **b** Spatial illuminance across model floor space at various fluid injection levels. **c, d** Renders within model space at various fluid injection levels, where (**c**) shows RGB-scale render and (**d**) shows false-color render based on illuminance. **e** Transmission spectra of available optical states for control algorithm to switch between along south facade for EC window, RS, and fluid cell systems. **f** Simulated cumulative solar heat gain, representing the sum of direct and diffuse solar ingress, within the model space across the year when clad with the fluid cell, EC window, RS, and static control window. **g–i** Simulated cumulative heating, cooling, and electric lighting energy usage within the model space across the year when clad with the fluid cell, EC window, RS, and static control window. **j** Total simulated cumulative energy usage within the model space across the year when clad with the fluid cell, EC window, RS, and static control window.

from a 1D reservoir to 2D area, while organized into independent cells within an array. Analogous to the krill, a small volume of stored pigment fluid (20 mL) can change the overall light transmission of a $30 \times 30$ cm$^2$ area by over 90% when injected and expanded, without an energy requirement to maintain the absorption (shaded) state. While these dynamic optical transitions are much slower than conventional digital displays (seconds to minutes, rather than the <10 ms response of a typical LCD

display), they fit well within the necessary response time of active building facades in changing solar conditions.

Importantly, buildings are among our civilization's costliest energy sinks, consuming approximately 75% of the electricity in the United States alone[15]. Any objective to reduce energy efficiencies and total carbon emissions globally should immediately recognize the need for even modest improvements in building design. Despite significant improvements in energy efficiency in certain technologies (e.g., transportation, energy-harvesting), building efficiency improvements have progressed only moderately within the past century, and in many respects has even declined. For instance, while the energy costs and inefficiencies of glass windows have been recognized since the mid-19th century, our use of glazing in buildings, and the associated indoor heating and cooling costs, has significantly increased[14,78].

Buildings with significant glazing ratios must compromise between maximizing total indoor daylight illumination, while limiting localized glare, while balancing the impact of solar radiation on heating and cooling costs year-round. The development of building materials that can find this compromise – i.e., that can actively shade by locally toggling between the optical performance of a window and wall – might simultaneously increase total illumination, reduce concentrated glare and discomfort, and substantially lower mechanical heating and cooling requirements[79]. This work represents significant progress in this direction, as we achieve locally-responsive shading that might overcome the functional limitations of more traditional macro-scale mechanical mechanisms (e.g., blinds).

Moreover, because the optical properties and functions of fluids can be easily tuned, switchable fluid expansion can be leveraged to control a range of responses beyond binary opaque shading – for example, directionally-programmable light scattering, polarization, and spectrally-selective absorption of IR light (this would crucially decouple control of daylight and heat gain indoors, which is another fundamental challenge in building design). Using a pigment phase that can partially transmit incident light might also be more desirable in building applications.

In addition to area fraction control of injected pigment fluid, we leverage an interfacial branching instability to tune morphology. Branching, as opposed to stably expanding[80], pigment coverage enables optimal length-scales of shading for minimal volumes of fluid. Krill achieve branching morphologies through a fixed channel structure, while, in this work, we tune branching dynamically through active control of injection flow rate. Over large areas, the degree of branching can be specified to satisfy a desired shading response, through a minimal, optimized, volume of pigment. Nonetheless, we found that branching morphologies were unstable over time, suggested better temporal control of shading with non-branching fluid configurations.

Finally, dynamic control over multiple fluidic cells enables highly-localized, digitally-programmable shading responses. Digital control importantly ensures that a whole building response can be optimized for maximum energy efficiency in varied hourly, diurnal, and seasonal environmental conditions. In this vein, we showed that fluid reconfigurations at hourly time-steps could achieve massive performance improvements, saving more than 30% on annual heating, cooling, and lighting energy compared to a state-of-the-art electrochromic window. With this fluidic control established, we imagine that artificial intelligence algorithms can collect, processes, and act upon large amounts of localized environmental data, even more drastically improving system management and energy efficiency. Ultimately, there is great potential for digitally-controlled, active shading to allow the next generation of buildings to learn, with

fundamental implications for an architecture that designs and redesigns itself.

## Methods

**Prototype fabrication and preparation.** Several prototype devices were fabricated based on the designs described in Ref. [69], ranging in surface area from $5 \times 5\ cm^2$ to $45 \times 45\ cm^2$. All prototypes were fabricated as PMMA-liquid-PMMA sandwiches (Hele-Shaw cells, 1-mm fluid gap). For single-cell devices, an inlet hole (5.5 mm diameter) was drilled into one of the PMMA plates, and a luer adapter with a barbed hose fitting was sealed to the inlet using two-part epoxy resin. PVC tubing (1/4" I.D., 3/8" O.D.) was connected to a digital peristaltic pump (INTLLAB, RS385-635) or digital syringe pump (New Era Pump Systems, NE-1010) for experimentation. The space between plates (which were assumed to be rigid through pigment injection) was sealed with a 1-mm-thick sheet of double-sided tape (3M), which acted as a durable and waterproof boundary for the enclosed liquid layer. An outlet hole (5.5 mm diameter) was drilled to allow air to escape during filling.

The devices were first filled with transparent host liquid (oil). Experiments were conducted by pumping pigment fluids into the oil-filled cell. If both plates of the cell were rigid (3-mm-thick PMMA), four outlets were established (one at each corner of the cell, 5.5 mm diameter). Each outlet enabled a small volume of liquid to reversibly leak and return into a container open to the atmosphere. This method was used for pattern morphology testing, as plate thickness could be kept relatively constant (plate thickness impacts branching pattern features). For multi-cell prototypes, a thinner PMMA plate (0.2 mm thick) was used to alleviate the need for outlets. All PMMA sheets were transparent, however a white sheet of plastic was placed directly behind the cell to observe fluidic patterning more clearly. All PMMA sheets were either milled using a three-axis CNC-mill (AXYZ Pacer 4010 ATC) or cut with a laser cutter (Universal Laser Systems PLS 6.150D). The tape gasket was cut manually.

**Fluid preparation and viscosity measurements.** We used castor mineral oil (Heritage Store) as the clear host liquid. We used different glycerol-water (BioShop, purity 99%) solutions with suspended carbon black particles (Davis Colors, 0.2 g C per 50 mL glycerol-water solution) as the pigmented liquid. Mixtures were sonicated (iSonic D3200) for 120 s. Viscosities of glycerol-water solutions were calculated using the four-parameter correlation of temperature dependence on aqueous glycerol solution viscosities, presented by Chen and Pearlstein[81], and were further verified by comparing our calculated values to experimental values measured by Segur and Oberstar[82]. The viscosity of the carbon suspension in water was measured using a Cannon-Fenske capillary viscometer (Sigma–Aldrich Z275301), and was found to be identical to water by itself (1 cP). The viscosity of castor oil was also measured using the viscometer (288 cP), and broadly confirmed with[83].

**Density matching experiments.** To eliminate buoyancy differences, we used a water/ethanol solution (23 vol%) as a guest fluid to match the density of the host fluid (castor oil) at 23 °C (0.95 g/mL). Mixing of these miscible liquids does not result in a significant change in partial molar volumes[84]. Experiments with vertical devices confirmed there were no drifting effects observed over multiple-hour-long periods.

**Branching pattern characterization and flow rate measurements.** Branching fluidic patterns were characterized based on the number and thickness of branches formed for various flow rates. Fluid area coverage was calculated in ImageJ (NIH, United States). The number of branches were counted and marked in Rhinoceros3D (McNeel, United States). Radius was measured based on a circle that fully enclosed all branching features. Pattern perimeter was measured digitally in Rhinoceros3D. Characteristic wavelength was determined based on the thickness of the most unstable wavelength – that is half of the width of a finger branch at the moment before it begins to split, as described in previous work[65]. For viscosity tests, an inner and outer circle was defined, respectively, as a circle that completely enclosed the inner fluidic area, and as a circle that completely enclosed all fluidic features. These were identified and defined manually, and their radii, perimeters, and areas were measured digitally in Rhinoceros3D. Flow was generated and measured using a NE-1010 digital syringe pump. A syringe was connected to inlet PVC tubing (1/4" I.D., 3/8" O.D.), which connected the pressurized syringe to the cell.

**Light intensity measurements and electronic feedback.** We programmed an Arduino MEGA 2560 R3 (Elegoo) to translate the output of a simple photosensor (Adafruit 161) into a proportional input for a 12 V DC digital peristaltic pump (INTLLAB RS385-635). We used a handheld LED light source (Neewer 10095736) to provide a constant light intensity of 100 lux. For single-sensor experiments, we used an Extech HD450 Light Meter Datalogger, placed behind our cell, to measure and log light intensity. Light intensity reduction values were calculated as percentage reductions, by taking the difference in visible light intensity (measured behind the cell) prior to, and after, complete fluid injection, and dividing this number by the maximum visible light intensity measured prior to injection.

**Temperature measurements and electronic feedback**. We programmed an Arduino MEGA 2560 R3 (Elegoo) to translate the output of a digital K-type thermocouple (HiLetGo) into a proportional input for a 12 V DC digital peristaltic pump (INTLLAB RS385-635). The experimental setup is detailed in Fig. 3k. We used an incandescent light bulb as a heat source that generated a constant power of 100 W. We used a K-type thermocouple (0.523 kJ/kgK) to measure the temperature of a PMMA sheet (1.42 kJ/kgK), 3 cm behind the fluidic device.

**Optical spectral measurements**. UV-vis-infrared spectrophotometry (Perkin-Elmer Lambda 1050) was performed for both clear and pigment fluids.

**Local and differential light intensity control and electronic feedback**. We connected one digital peristaltic pump (INTLLAB RS385-635) to the inlet tubing for every cell, and placed a photosensor (Adafruit 161) 2 cm behind each cell to measure cell-specific local light intensity. We applied a similar control algorithm as described for individual cells, and illuminated individual cells in sequence to generate a fluidic pigment response to independent local light intensity changes.

**Simulated optofluidic displays**. A Python program was developed to first input and convert RGB images as greyscale multipixel arrays, next average regional collections of greyscale pixels, and finally replace multipixel collections with experimental images of fluid injection. We replaced greyscale pixels with images of our fluid injection experiment. For smaller pixel greyscale values (darker), we replaced the pixel with an image of a fluid injection after proportionally longer runtimes (i.e., more fluid displaced, larger pigment pattern). In particular, we sorted the pixels into 5 buckets and proportionally matched up the pixels with experimental images of varying degrees of injection. As a result, we computationally generated several half-tone displays using experimental images of stable, quasi-circular, injection sequences.

**Building energy simulation control algorithm**. At every hour that the modelled space was occupied, each dynamic control state (Fig. 6e) was tested for the preset daylighting requirements (illuminance of 300 lux across at least half of the floor area, and an illuminance above 3000 lux across less than 10% of the floor area). For every state that met these requirements, electric lighting utilization and solar heat gains were calculated, followed by a heat balance at each hour based on model outputs for internal heat gains (occupants, lights, equipment) and external heat gains (ventilation, solar heat gains, conduction). At each hour, of all the tested states, the state that minimized total heating, cooling, and electric lighting energy was selected. Finally, to account for the transient nature of the thermal model, the control algorithm was iterated many times until the annual energy results stabilized at a near-optimal solution.

**Simulation parameters**. EnergyPlus was used to simulate annual energy usage within a previously defined commercial reference space, described in[72]. A standard hourly occupancy schedule was assumed for the space[85], with an occupancy density of $0.0538$ m$^2$/person. Non-exterior walls, floors, and ceilings were set as adiabatic. Opaque walls were modelled with a U-value of $0.472$ W/m$^2$K. A fresh air supply of $0.0125$ m$^3$/s/person was used, and we assumed that 70% of sensible heat and 65% of latent heat was recovered by the heat-recovery system. Each person generated 125 W of heat, and indoor equipment was modelled with a maximum power density of 5 W/m$^2$. An hourly schedule defined by the National Energy Code of Canada for Buildings[85] was used for occupancy, equipment, and temperature setpoints. Such temperature setpoints were defined at 21 and 24 °C when the space was occupied, and at 15.6 and 26.7 °C otherwise. Heating system efficiency was assumed to be 80%, and the air conditioning system was assigned a coefficient of performance of 3.2. An electric lighting power of 99 W (3.4 W/m$^2$) across the space was assumed. Finally, lights were activated daily between 7:00–19:00, and were dimmed to meet a target illuminance while optimizing thermal gains.

## Data availability
All data generated in this study can be found in this manuscript and Supplementary Information file.

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

## Acknowledgements

We acknowledge the help of Ethan Heimlich for developing the fluidic half-tone visualization algorithm. We also thank Nicholas Hoban for his conceptual and technical input early on in the project. This work was supported by the Canadian Foundation for Innovation #31799 (B.D.H.), a Percy Edward Hart Professorship, University of Toronto (B.D.H.), and the Climate Positive Energy program, University of Toronto (B.D.H.). R.K. acknowledges support from a Canada Graduate Scholarship, C.W. Bowman Graduate Scholarship, Bert Wasmund Graduate Fellowship, and Hatch Graduate Scholarship.

## Author contributions

Conceptualization: R.K., C.K., B.D.H. Methodology: R.K., C.K., K.N., B.D.H. Physical experimentation: R.K. Simulation and control algorithm design: J.A.J. Visualization: R.K. Electronic system design: K.N. Funding acquisition: B.D.H. Initial draft writing: R.K. Editing and revision: R.K., C.K., K.N., J.A.J., B.D.H.

## Competing interests

The authors declare no competing interests.
