## [Peer Review File · Nature Communications]

Decapod-inspired pigment modulation for active building facadesREVIEWER COMMENTS

Reviewer #1 (Remarks to the Author):

This is a classic paper with clearly identifies objectives, specific scope, well documented and presented results and straight forward conclusions. The authors study the potentiality of confined layers enclosing pigmented fluids for solar control in buildings. The authors' efforts resulted in the design, construction and testing of a multi-cell system driven by injection/extraction devices of pigmented fluids

The reviewer thinks that the questions addressed in this paper are novel and original and deserve the interest of the scientific community involved in the field of solar control in buildings. The paper is reasonably well written and clear. The principle of operation and the experimental set-up is described with good detail. The conclusions formulated by the authors seem to be correctly drawn from the combined simulation and experimental results. The reviewer retains that the technical information presented in the paper may be of importance for the design and engineering implementation of solar control systems in buildings for energy saving purposes.

Therefore, the reviewer recommends publication of the paper, only advising the authors to consider the few minor corrections listed below:

- All parameters appearing in equations 1-3 depend on temperature. Since a window is subjected to daily and seasonal temperature excursions, which is the influence of temperature on the performance and control of the system developed by the authors?
- The authors should indicate if they expect that the performance of the system degrade with the number of injection/extraction cycles. If yes, can they estimate the durability of their device?
- When estimating the energy saving deriving from the building integration of their component the authors should consider, in addition to the cooling and heating energy consumption, also the lighting energy consumption. This could modify substantially the real energy saving obtainable.
- An important feature of electrochromic windows is the ability to maintain visual contact between indoors and outdoors in almost all transmitting states. The system proposed by the authors seem instead to loose, or at least, to change visual contact at increasing injection states. This fact could be not favourably accepted by users. Please, can you add some comments on this issue?
- In the introduction the authors affirm that current dynamic windows are "expensive and complex to manufacture". Can the authors estimate the fabrication cost of their (full-scale) system and discuss briefly the fabrication complexity level? Furthermore, which is the energy supply required compared to an electrochromic window?

Reviewer #2 (Remarks to the Author):

The topics covered are certainly interesting and innovative, and open up new horizons for research and development in the building sector.

While appreciating the theoretical discussion, however, there are some doubts about the experimental and applicative aspects. There are also some critical points in the text.

Below are some comments, first on the specific contents and then of a general nature.

The threshold of 3 kW/m² for solar irradiation is unreasonable, since the limit value on the earth's surface is 1, with rare exceptions towards 1.1. Even taking into account multiple reflections from the surroundings, the value proves to be excessive. It may be an oversight, but it is certainly not an effective way to start a dissertation.

Experimental equipment is described in the text, but the pictures only show three-dimensional drawings. It would be useful to show the actual equipment as well.

The threshold of 100 lux used in the experiments seems inadequate, as solar radiation can cause

illuminance of up to 100,000 lux and more.

On lines 254-257 the comparison in terms of energy savings is unclear and so the reference to Figure 6.

Low-cost technology is mentioned, but no economic evaluation is reported.

In general, the comparison with systems currently available on the market is approximate.

Shading systems for glazed façades are widespread and effective, including adjustable Venetian blinds, movable slats, double-glazed lamellas, etc. Limiting the comparison to just fixed blades is misleading.

In the reviewer's opinion, one important issue was overlooked. Theoretically and after making the suggested corrections, the research can be important. Its field of application, however, is the construction industry, where economic, environmental and functional boundary conditions are crucial.

An assessment should be made of the effectiveness and durability of the technology developed under real operating conditions, taking into account the prolonged action of weather conditions as well as construction and regulatory constraints.

Finally, a thorough techno-economic assessment, also in relation to competing technologies and achievable energy savings, should not be missing, including manufacturing, installation, maintenance costs and estimates of the useful life of components.

Reviewer 1

- 1. This is a classic paper with clearly identifies objectives, specific scope, well documented and presented results and straight forward conclusions...The reviewer thinks that the questions addressed in this paper are novel and original and deserve the interest of the scientific community involved in the field of solar control in buildings...Therefore, the reviewer recommends publication of the paper, only advising the authors to consider the few minor corrections listed below.*

We thank Reviewer 1 for their assessment and insightful comments.

- 2. All parameters appearing in equations 1-3 depend on temperature. Since a window is subjected to daily and seasonal temperature excursions, which is the influence of temperature on the performance and control of the system developed by the authors?*

We agree that these are important considerations, and have demonstrated how our system can respond to, and perform under, regular fluctuations in outdoor temperature. Primarily, it is the viscosities of the host and injected fluids that will vary with temperature and, in compensation, the injection flow rate can be adjusted accordingly.

We have used a conduction finite difference solution algorithm within our energy model to estimate the temperature fluctuation of our fluid layer at all points of the year in Toronto, Canada, when modelled on the interior of a double-glazed window. The most extreme temperatures experienced by our fluids (minimum of 8.9 and maximum of 44.3 °C, while on the protected interior of the double-glazed window) in the highly temperature-variable climate of Toronto, Canada result in changes in viscosity difference of 57% and 316%, respectively, compared with the viscosity difference calculated at laboratory temperatures (20.0 °C) (all calculations shown in Operational Temperature Dependence section in the SI and tabulated in Extended Data Table 1).

Using equations (2-3), we show that the estimated change in injection velocity to compensate for minimum and maximum temperature deviations from laboratory conditions is within an order of magnitude, and well within the functional range of our peristaltic pumps. Specifically, we calculate that at the coldest modelled annual temperature, the injection velocity would need to be lowered to 43% of its laboratory value, while, at the hottest modelled annual temperature, the injection velocity would need to be raised to 415% of its laboratory value, to combat such temperature fluctuations.

Therefore, this estimated change in the relative viscosity should be within the design control of our system. Temperature therefore must be considered by a digital control algorithm, but by no means limits the range of pattern control demonstrated in this paper. Additionally, the host and injection fluids can themselves be chosen to have suitable viscosities (and $d\eta/dT$) for a given climate.

- 3. The authors should indicate if they expect that the performance of the system degrade with the number of injection/extraction cycles. If yes, can they estimate the durability of their device?*

This is an important point, and we have done significant experimental work to address the repeatability and durability of fluid injections. An entire section within our paper (“Reversibility of Injections and Switchable Injection Stability”) is largely dedicated to identifying the range of fluid parameters necessary to achieve stable, reversible, and repeatable fluid injections within our vertical device. As these fluids are immiscible, and stable against oxidation, the main mechanism for ‘degradation’ with repeated

cycling is when highly branched structures do not fully retract, and leave behind discrete fluid droplets over many injection cycles. The non-branched morphologies do not show this phenomenon, and for branched structures we have defined physical criteria to specifically avoid droplet formation from occurring.

We have shown that stable, non-branching injections, achieved using immiscible fluids with no difference in viscosity (288 cP for both oil and aqueous phases), were repeatable over 100 cycles, showing no signs of degradation. This result has been added to the “Reversibility of Injections and Switchable Injection Stability” section.

- 4. When estimating the energy saving deriving from the building integration of their component the authors should consider, in addition to the cooling and heating energy consumption, also the lighting energy consumption. This could modify substantially the real energy saving obtainable.*

We completely agree with this comment, and, accordingly, we have added an additional co-author (J. Jakubiec) who is an expert in building energy simulations. We have added a more thorough building energy model, which accounts for light energy consumption. Light energy was calculated using a well-established backwards light-ray-tracing simulator and renderer (Radiance), given time-stepped luminance values calculated with six ambient bounces, for seasonal incident solar data. Its results are discussed and demonstrated in Fig. 6, and we now report that our fluid system can reduce heating, cooling, and lighting energy by over 30%, compared to the state-of-the-art electrochromic technology.

- 5. An important feature of electrochromic windows is the ability to maintain visual contact between indoors and outdoors in almost all transmitting states. The system proposed by the authors seem instead to lose, or at least, to change visual contact at increasing injection states. This fact could be not favourably accepted by users. Please, can you add some comments on this issue?*

We agree with this point, and have added a comment in the discussion section. In this paper, we have showed control over the scale of fluid injections, and the area fraction of complete solar shading (i.e., degree of visual contact). It is important to note that our fluid mechanism is generalizable, and can be applied using guest fluids with lower concentrations of pigment and with a range of optical properties (e.g., cloudy suspensions, partially-transmitting pigment phases), which is part of our on-going work. Depending on the design requirements, we speculate that only partially transmitting fluids can be used, to maintain better visual contact across the facade.

- 6. In the introduction the authors affirm that current dynamic windows are “expensive and complex to manufacture”. Can the authors estimate the fabrication cost of their (full-scale) system and discuss briefly the fabrication complexity level? Furthermore, which is the energy supply required compared to an electrochromic window?*

Our system has been designed to consist of relatively inexpensive materials (PMMA sheets, castor oil, water, carbon black, and off-the-shelf electronics), and we estimate the cost per square meter to be roughly \$40 USD. This compares favourably to the \$500 USD/m² cost of electrochromic windows. Fabrication requires PMMA sheet cutting and adhesive assembly (Methods), and material costs consist primarily of the PMMA sheets and electronic setup (many more details and numbers in the Technoeconomic Feasibility section).

Regarding the latter point, we have developed a comparative energy cost estimate for operating both our dynamic fluid system and a dynamic electrochromic window (latter from the literature). We found that our estimated operational energy cost of 0.15 kWh/m² is only 30% of a typical operational energy cost estimate of an EC window (0.5 kWh/m²), and 45 times less than the operational energy cost of a device that requires a constant low-power energy supply (~5 W/m²) to shade (e.g., liquid-crystal and suspended-particle devices, 6.82 kWh/m²).

Additionally, we calculated – based on estimated implementation costs, estimated energy savings, and the current cost of energy – a payback time for the costs associated with cladding our reference model (from simulation) along its south facade in our fluidic system (1.24 years, or about 15 months).

Reviewer 2

- 1. The topics covered are certainly interesting and innovative, and open up new horizons for research and development in the building sector. While appreciating the theoretical discussion, however, there are some doubts about the experimental and applicative aspects. There are also some critical points in the text.*

We thank Reviewer 2 for their assessment and insightful comments.

- 2. The threshold of 3 kW/m² for solar irradiation is unreasonable, since the limit value on the earth's surface is 1, with rare exceptions towards 1.1. Even taking into account multiple reflections from the surroundings, the value proves to be excessive. It may be an oversight, but it is certainly not an effective way to start a dissertation.*

We appreciate this correction. We have used solar irradiation values from our energy model to provide an accurate number, where we calculated instantaneous peak solar irradiance on a vertical surface in Dubai, using climate data averaged across the past twelve years, to find a maximum value of 832 W/m². We have corrected this number accordingly in the text.

- 3. Experimental equipment is described in the text, but the pictures only show three-dimensional drawings. It would be useful to show the actual equipment as well.*

We have added an Extended Data figure (13) that shows various aspects of our experimental setups.

- 4. The threshold of 100 lux used in the experiments seems inadequate, as solar radiation can cause illuminance of up to 100,000 lux and more.*

Because the transmission spectra of our fluids are independent of the absolute value of incident illumination, only the transmissivity (i.e., transmission fraction) value in our experiments is relevant to our ultimate optical and energetic performance. Accordingly, in Fig. 21-n, our light transmission data is now clearly defined as percent transmission data, generalizable to incident illuminance of 100,000 lux. We also note that in our energy model, real-life historically-averaged solar radiation illuminance is used, and the performance values of our system are based on these solar-illuminance values.

- 5. In general, the comparison with systems currently available on the market is approximate. Shading systems for glazed façades are widespread and effective, including adjustable Venetian blinds, movable slats, double-glazed lamellas, etc. Limiting the comparison to just fixed blades is misleading.*

We completely agree that our energetic analysis should include other existing technologies, and be much more quantitative in nature.

To address this, we have brought on board an additional co-author (J. Jakubiec) who is an expert in building energy simulations. We have developed a much more thorough building energy model which accounts for light energy consumption, and we have developed a new control algorithm for comparing multiple dynamic systems.

In our newly implemented model, we have calculated the annual energy required to heat, cool, and light a standard reference space when clad not only in our dynamic fluid system, but with a dynamic electrochromic window, a static window, and a static window with an adjustable roller shade. In particular, we found that the dynamic operation of our fluid cell reduced total annual operational energy usage within our model space (the sum of heating, cooling, and electric lighting energy usage) by 30.1%, 22.9%, and 20.4%, compared to electrochromic, roller shade, and static double-glazing systems, respectively. These new energy results provide a specific quantitative comparison between operational energy savings provided by available chromogenic building technologies.

Moreover, we have compared the estimated energy cost to operate both our dynamic fluid system and a dynamic electrochromic window, estimating that our system requires only 30% the energy needed to operate an EC window, and 1/45th the energy needed to operate a dynamic LCD window that requires continuous power to shade.

6. On lines 254-257 the comparison in terms of energy savings is unclear and so the reference to Figure 6.

This section and figure have been replaced with our more thorough energy model.

7. In the reviewer's opinion, one important issue was overlooked. Theoretically and after making the suggested corrections, the research can be important. Its field of application, however, is the construction industry, where economic, environmental and functional boundary conditions are crucial. An assessment should be made of the effectiveness and durability of the technology developed under real operating conditions, taking into account the prolonged action of weather conditions as well as construction and regulatory constraints...

We greatly appreciate this suggestion. Accordingly, as we mention above, we have done significant experimental work to assess the repeatability and durability of fluid injection/withdrawal cycles within our system (response to Reviewer 1, point 3).

The “Reversibility of Injections and Switchable Injection Stability” section within our paper is, in fact, largely dedicated to identifying the range of fluid parameters necessary to achieve stable, reversible, and repeatable fluid injections within our vertical device, with the objective of system durability in mind. In this section, we discuss how the main mechanism for ‘degradation’ with repeated cycling is when branched structures do not fully retract, and leave behind discrete fluid droplets over several injection/retraction cycles. The non-branched morphologies do not show this phenomenon, and for branched structures we have defined physical criteria to specifically avoid droplet formation from occurring.

In particular, we have added results which validate that such stable, non-branching, injections, achieved using immiscible fluids with an equivalent viscosity, are repeatable over 100 cycles, showing no signs of degradation. Because of this thermodynamic stability, we are able to estimate that the durability of this mechanism, in terms of number of repeatable fluid injection and retraction cycles, is non-limiting in terms of the overall robustness of the material. This result is added to the “Reversibility of Injections and Switchable Injection Stability” section.

We also appreciate the concern of the system’s effectiveness in our laboratory environment versus in real operating conditions, where weather and temperature fluctuations are a factor. Accordingly, we have added discussion about the construction integration of our fluid layer within a conventional internal glazing unit, and we have commented on the deliberate placement of our fluid layer on the inside of the insulating portion of a double-pane window.

More so, we have added a new series of calculations to assess how the viscosities of the host and injected fluids that will vary with temperature and, in compensation, how the injection flow rate can be adjusted accordingly (which are described in the response to Reviewer 1, point 2). A conduction finite difference solution algorithm within our energy model was used to estimate the temperature fluctuation of our fluid layer at all points of the year in Toronto, Canada (when modelled on the interior of a double-glazed window). The most extreme temperatures experienced by our fluids result in changes in viscosity difference of 57% and 316%, respectively, compared with the viscosity difference calculated at laboratory temperatures (20.0 °C). These calculations are shown in the Operational Temperature Dependence section in the SI and are tabulated in Extended Data Table 1. Using equations (2-3), we then showed that the change in injection velocity to compensate for minimum and maximum temperature deviations in real-life operating conditions is within the functional range of our peristaltic pumps.

Finally, the placement of our fluid layer on the interior side of a double-glazed window can provide weatherproofing and durability/sheltering from the outdoor environment.

8. Finally, a thorough techno-economic assessment, also in relation to competing technologies and achievable energy savings, should not be missing, including manufacturing, installation, maintenance costs and estimates of the useful life of components...Low-cost technology is mentioned, but no economic evaluation is reported.

We certainly agree that a more comprehensive economic evaluation would benefit this paper, and a “Technoeconomic Feasibility” section has been added.

Within it, we have provided a quantitative estimate of the costs associated with our system’s annual operation (0.15 kWh/m²). More, we have compared this estimated cost with an estimate of the energy cost to operate a dynamic electrochromic window (0.5 kWh/m²), 2.3 times more, and an estimate of energy cost to operate a dynamic LCD window that requires continuous power to shade (6.82 kWh/m²), 45 times more.

We have also compared these operational energy costs to the absolute energy reduction between our fluid system and an electrochromic system, to provide important context about the relative difference in energetic costs. We found that the operational energy cost of our fluid layer is less than 0.25% of the total heating, cooling, and lighting energy reduced by our fluid layer over an EC window (578.8 kWh).

Moreover, we have provided a comparative estimate for the energy required to manufacture both our dynamic fluid system plus glazing (~ \$200 USD/m²) and a dynamic electrochromic window (\$500-1000 USD/m²), and we have commented on the longevity and durability of our system's solid and fluid components.

Critically, our system consists of relatively inexpensive and readily-accessible materials (PMMA, castor oil, water, carbon black, off-the-shelf electronics), and the cost of our system by itself, per square meter, is roughly \$40 USD. Fabrication avoids any complex synthesis or microelectronic processing conditions, requiring only PMMA sheet cutting and adhesive assembly (Methods). Material costs consist primarily of the PMMA sheets and electronic setup (added to Technoeconomic Feasibility section), while we expect maintenance to be relatively easy (replacement of pigmented liquid) (added to Technoeconomic Feasibility section, and compared against degradation concerns prevalent in electrochromic windows).

Finally, we conclude the section by calculating, based on estimated implementation costs, estimated energy savings, and the current cost of energy, a payback time for the costs associated with cladding our reference model (in simulation) along its south facade in our fluidic system (1.24 years or 15 months).

This addition provides a more thorough embedded techno-economic assessment to accompany the above operational energy saving and cost comparison. We also emphasize that the introduced principle of confined fluid control is particularly attractive due to its low fabrication/processing/materials/operational costs (Technoeconomic Feasibility section).

Reviewer 3

We thank Reviewer 3 for their assessment and insightful comments.

1. *How is the use of 'self-organized' in the title justified? The injection of fluids and viscosity contrast results in different patterns. The fluids just don't organize themselves. Such a title may mislead.*

The viscous fingering mechanism for fluids in confined volumes is a well-established example of self-organized pattern formation – see any of References 62-68 within our paper, or, externally, any of: Chen J, Qin S, Wu X, et al. Morphology and Pattern Control of Diphenylalanine Self-Assembly via Evaporative Dewetting. *ACS Nano* 2016; 10: 832-838; Kuroda A, Ishihara T, Takeshige H, et al. Fabrication of Spatially Periodic Double Roughness Structures by Directional Viscous Fingering and Spinodal Dewetting for Water-Repellent Surfaces. *The Journal of Physical Chemistry B* 2008; 112: 1163-1169; Qin, S., Yan, J., & Wu, J. (2016). Self-growing features of viscous fingering process modelled by DLA. *Management & Engineering*, (22), 3-6.

The fractal patterns generated through viscous fingering are self-similar. This spontaneous order comes from internal interactions of the system, where fingers are self-avoiding, and so self-organizing into a branched network, which in our case shades the cell area by varying fractions for different injection parameters.

2. *What is the duration of injection and suction? Both are done for the same duration of time? Does this duration have any effect on the light intensity?*

Depending on injection flow rate, injection and suction processes take between 15 s and 24 m – representing a timescale that is responsive to the needs of a building.

We have reported the starting and finishing time stamps of video frames shown in Fig. 2f. And, as we demonstrated in the time-dependent graphs in Fig. 2l-n, we note that injection and suction stages take the same time for each flow rate sequence. We also note that light intensity as a function of time is demonstrated in Fig. 2l-n, and a better picture of fluid coverage (which is linearly related to light transmission/intensity) is shown as a function of time for all injection and suction sequences in Fig. 2g. Time is consistently reported for data (e.g., Fig. 3e-f, i-j, m-n, Fig. 4, Extended Data Fig. 2d), although, we have reported more temporal data for a few images with missing time stamps (e.g., Fig. 5c), for completeness.

3. How is retraction performed as there is an outlet to allow the fluid to flow out?

Retraction is performed as the reverse of injection, as the inlet flow is reversed. This is demonstrated in particular in movies S1-S5.

4. The light can increase the temperature which can change the viscosity of the liquids. What will be the effect on VF and hence the light may be discussed.

We appreciate this important point. Both the absorbed incident sunlight and exterior temperature fluctuations indeed will change the temperature of the fluid layer.

In response to this point, we have demonstrated how our system can perform under regular fluctuations in outdoor temperature and solar absorption (also see the similar response to Reviewer 1, point 2 and Reviewer 2, point 7).

The viscosities of the host and injected fluids will vary with temperature. We demonstrate how, in compensation, the injection flow rate can be adjusted accordingly.

We have used a conduction finite difference solution algorithm within our energy model to estimate the temperature fluctuation of our fluid layer at all points of the year in Toronto, Canada, when modelled on the interior of a double-glazed window. The most extreme temperatures experienced by our fluids (minimum of 8.9 and maximum of 44.3 °C, while on the protected interior of the double-glazed window) result in changes in viscosity difference of 57% and 316%, respectively, compared with the viscosity difference calculated at laboratory temperatures (20.0 °C) (all calculations shown in Operational Temperature Dependence section in the SI and tabulated in Extended Data Table 1).

Using equations (2-3), we show that the estimated change in injection velocity to compensate for minimum and maximum temperature deviations from laboratory conditions is within an order of magnitude, and are within the functional range of our peristaltic pumps. Specifically, we calculate that at the coldest modelled annual temperature, the injection velocity would need to be lowered to 43% of its laboratory value, while, at the hottest modelled annual temperature, the injection velocity would need to be raised to 415% of its laboratory value, to compensate for such temperature fluctuations.

Accordingly, this estimated change in the relative viscosity should be within the design control of our system: appropriately tuning the flow rate should compensate for any temperature-dependent changes in pattern stability. Temperature therefore must be considered by a digital control algorithm, but by no means limits the range of pattern control demonstrated in this paper. Additionally, the host and injection fluids can themselves be chosen to have suitable viscosities (and $d\eta/dT$) for a given climate. We expect to scale up these systems and test in outdoor environments in our future work, to test this point.

I wonder how each site in a multicell are stopped to come in contact, and if VF occurs at each site, then how it affects the glare control and energy saving in a multicell system, as only stable fronts are discussed in figure 5.

This first point was certainly experimented on. We found that for a multicell system with no physical cell boundary, the injected fluids tended to intermix, causing issues with reversibility and repeatability. That is why, for our multicell systems, cells are physically separated by a transparent double-sided adhesive gasket, the design of which is described in Methods.

Regarding the second point, we have added a new author (J. Jakubiec) to help implement a light distribution analysis using a well-established backward-ray tracer simulator to compare indoor illuminance for different VF patterns that were demonstrated in experiment. In Extended Data Fig. 12, we show the effect of branching on average illuminance, over-lit area (glare), and useful daylight utilization (related to lighting energy requirement).

We note that the energy saving in a multicell system is a function of the number of possible fluid states achievable for controlling incident light transmission. Because the range of control over light intensity achieved through different branching patterns is encompassed completely within the range of control over light intensity achieved through different pattern sizes, the occlusion or inclusion of branching patterns within or model has no effect on the total heating/cooling/lighting energy balance possible when optimizing our system.

5. The VF in fluid facades is shown to more effectively modulate the light emission by 24% compared with a circular stable displacement when the injection site is one. But the multiple site case with VF, no such modulation on large displays is shown, which could be interesting aspects to present.

By controlling the morphology of fluid injections using VF, we showed that we could control the maximum light transmission through the cell by 24% - where the circular stable displacement represents the condition where the maximum light intensity reduction could occur. The circular stable state accordingly represents the state where the maximum range of light intensity modulation can occur, because it affords the maximum range of area coverage modulation, and represents the basis for why we chose to demonstrate stable injections for multicell displays. We do acknowledge the possible difference in light distribution (and glare/overlighting) for given VF conditions, and we therefore present newly simulated results to demonstrate the effect of branching on average illuminance, over-lit area (glare), and useful daylight utilization (related to lighting energy requirement) (Extended Data Fig. 12).

6. In line # 173: the light intensity 91 %, 80%, 67% for the different flow rates is shown, how are these values calculated?

From line 173: "As expected, transmitted light decreased as a function of pigment fluid area (Extended Data Fig. 7), to decrease interior visible light intensity by 91%, 80% and 67% for maximum injections with flow rates of 0.5, 1.0, and 10.0 mL/min, respectively (Fig. 2I-n, respectively)."

These values were calculated as a percentage reduction, by taking the difference in visible light intensity (measured behind the cell) prior to, and after, complete fluid injection, and dividing this number by the maximum visible light intensity measured prior to injection (before – after / before).

We have clarified this point and have added more experimental details to Methods.

7. What is ρ_a , ρ_o in line 148?

The represent the density of the aqueous phase and density of the oil phase, respectively. We have clarified this in the text.

8. What is 'a' in line 153?

Here, a should have been written as a_λ , as introduced in Equation (1). We appreciate bringing attention to this detail. We have made this correction in the text.

9. How the light intensity is calculated is not very much clear from the paper? What do the curves in figure 2 signify?

Fig. 3c shows how light intensity was measured throughout the paper (i.e., behind the cell). Fig. 2l-m report these values. We have added details about experimental techniques and equipment in the Methods section.

10. The experimental result repeatability may be reported.

As mentioned in more detail in our response to Reviewer 2, point 7, our “Reversibility of Injections and Switchable Injection Stability” section is dedicated to identifying the range of fluid parameters necessary to achieve stable, reversible, and repeatable fluid injections within our vertical device, with the objective of system durability in mind. In this section, we discuss how non-branching injections are much more repeatable, as highly branched structures can break up into irreversible fluid droplets over dozens of injection cycles). In this section we accordingly defined physical criteria (fluid properties and device control properties) to specifically avoid this phenomenon from occurring, for the purposes of repeatability.

In particular, to provide a precise result, we have reported that stable, non-branching, injections, achieved using immiscible fluids with an identified range of fluid parameters, are repeatable over 100 cycles, showing no signs of degradation. Because of this thermodynamic stability, we are able to estimate that the durability of this mechanism, in terms of number of repeatable fluid injection and retraction cycles, is non-limiting in terms of the overall robustness of the material. This result is added to the “Reversibility of Injections and Switchable Injection Stability” section.

11. Line # 186: The concepts of the text “In buildings, the fraction of solar radiation that is transmitted into a building is captured by a solar heat gain” Some reference may be given to this aspect.

We have added a reference for the source of this equation.

12. What type of fluids in general (other than those given in the paper) can be used for this kind of application?

This is a very important question, and, within the “Reversibility of Injections and Switchable Injection Stability” section of our paper, we identified, both theoretically and experimentally, a phase space where we describe the necessary fluid parameters for achieving stable versus unstable fluid injections within a vertical Hele-Shaw cell. Importantly, the host and guest fluids must be immiscible, and,

depending on the desired stability of the injection, we show how the viscosity difference between fluids, interfacial surface tension, injection velocity, and cell dimensions are all related. The accompanying data shown in Extended Data Figs. 8-9 shows the broad generalizability of this mechanism, and provides a blueprint for other systems that can use a range of immiscible fluid pairs for achieving similar reversible performance in stable and unstable fluid regimes.

13. The branching patterns are shown at what time? There are no images of retraction, only the images when a small amount of fluid is left are shown. Please show some intermediate images too.

We have added timestamps to the data in Fig. 2f, where the injection and retraction times range from 15 sec to 24 min, depending on injection flow rate. Regarding retraction images, in Fig. 2f, we show several video frames during fluid retraction (an equal number as during injection), and we have also included all movies that show the complete injection and retraction sequences (e.g., movie S2-3).

14. What 'r' is used in the area calculation in extended figure 1? Does the data in extended figure 1e match with that of Bischofberger et al (reference 63).

The two different 'r' values used in EDFig. 1d-e are further clarified in the figure caption, referring to different portions of the injection radius. The data in EDFig. 1e follows a similar broad trend as the reported data from Bischofberger et al (reference 65), although controls on the experiments were different, and cannot be fully compared.

15. In the numerical simulations, what type of equations are used for the computation purpose is not clear.

We have added a new author (J. Jakubiec) with an expertise in building energy simulations to develop a more rigorous energy simulation using the EnergyPlus software. The EnergyPlus model we run uses the conduction finite difference solution algorithm, referenced in the paper, to calculate heating and cooling loads. Interior illuminance is calculated used a well-established and widely used backwards light-ray-tracing simulator and renderer called Radiance, with six ambient bounces, also referenced in the paper. More details on the numerical simulations have been added to Methods.

16. On page 21, what is meant by 'larger greyscale values replaced by proportionally larger fluid pattern'?

We replaced greyscale pixels with images of our fluid injection experiment. For smaller pixel greyscale values (closer to black), we replaced the pixel with an image of a fluid injection after a longer time (i.e., more fluid displaced). In particular, we sorted the pixels into 5 buckets and proportionally matched up the pixels with experimental images of varying degrees of injection. This detail is expanded upon and added to the Methods section.

17. Controlling VF by varying the viscosity ratio (or difference) and the flow rate are nothing new here as already a well-known phenomenon in the literature (eg Nature Communications 5, 5265, 2014, Physical Review Letters 115, 174501, 2015, J. Fluid Mech. (2020), vol. 884, A16.)

We certainly agree, and much of this literature is referenced. The novelty of our work is the application of this well-established and well-cited theoretical mechanism towards functional and active materials, with a shape that can be tuned and dynamically switched for control over optical properties. It is

important to note that the viscous fingering mechanism has never been applied to active optical control, let alone applied at the scale of a building facade. We believe the application and scalability of viscous fingering defines the novelty of our manuscript.

REVIEWERS' COMMENTS

Reviewer #2 (Remarks to the Author):

Comments and requests for additions were implemented by the authors, greatly improving the quality of the manuscript.

This is certainly an important study, so the additional work can only improve the outcome, as is evident from the final version.

In my opinion, in this form the manuscript is worthy of publication without further revision.

Reviewer #3 (Remarks to the Author):

Thanks for the revised version. The authors have tried to revise with utmost care. I am fine with the justifications. However, I am not convinced about the answer regarding the self-organizing of the fluids in the immiscible fluid system. The viscous fingering occurs when we inject the less viscous fluid to another higher viscous one, so self-organizing occurs only in the case of a specific fluid system where some chemical thermodynamic can play a role. Authors may see the paper (J. Fluid Mech. (2020), vol. 898, A11, Phys. Chem. Chem. Phys., 2021, 23, 10926) how the partially miscible fluid system can make the self-organizing of the fluids which are due to the purely another thermodynamic effects like the spinodal decomposition. If the self-organizing of the fluids are due to the optofluidic effects, it is understandable, The author may clarify these points.

With the above corrections, the paper can be accepted for publication in Nature Communications.

Reviewer 3

Thanks for the revised version. The authors have tried to revise with utmost care. I am fine with the justifications. However, I am not convinced about the answer regarding the self-organizing of the fluids in the immiscible fluid system. The viscous fingering occurs when we inject the less viscous fluid to another higher viscous one, so self-organizing occurs only in the case of a specific fluid system where some chemical thermodynamic can play a role. Authors may see the paper (J. Fluid Mech. (2020), vol. 898, A11, Phys. Chem. Chem. Phys., 2021, 23, 10926) how the partially miscible fluid system can make the self-organizing of the fluids which are due to the purely another thermodynamic effects like the spinodal decomposition. If the self-organizing of the fluids are due to the optofluidic effects, it is understandable, The author may clarify these points.

We would request that the title and text still include the term 'self-organized', as it's a very important aspect of what makes this work novel and impactful. Self-organization, in our context, refers to the hydrodynamic Saffman-Taylor instability (viscous fingering) that causes the highly branched morphologies we observe, as a function of flow rate (Fig. 2). One of our main points is that this branched, self-organized form is analogous to the branched pigment transport in organisms (krill). This morphology (VF branching) is a very well-known example of self-organization, in fact (discovered in the 1950s) and so it would be scientifically accurate and necessary to use this language. Additionally, the branching (self-organization) is key to our access to large areas with minimal fluid (pigment) volume. In our manuscript we use the terms 'self-organized', 'branched' and 'fluid instability' to all refer to this mechanism, so we suggest it is both important to the title, and well represented in the text of our manuscript.

We thank Reviewer 3 for their comments. Self-organization, in our work, refers specifically to the highly branched morphology of the Saffman-Taylor hydrodynamic fluid instability of viscous fingering (as pointed out, a liquid of low viscosity injected into a host fluid of high viscosity). For our system, the two fluids are highly immiscible. Therefore, we do not expect chemical thermodynamic effects, due to partial miscibility, to be active. In fact, our system is not at thermodynamic equilibrium, as you might find for other self-organized structures (such as reaction-diffusion, BZ patterns, or spinodal decomposition phase transformation).

However, we thank the reviewer for these references, as they show very interesting effects due to partial miscibility. And we expect to consider these 'chemical' mechanisms for future directions in our work, in addition to our 'physical' mechanism.